# The Influence of the Grassland Ecological Compensation Policy on Regional Herdsmen's Income and Its Gap: Evidence from Six Pastoralist Provinces in China

**Mengmeng Liu [1,2], Limin Bai [3], Hassan Saif Khan [1,2] and Hua Li [1,2,*]**

1   College of Economics and Management, Northwest A&F University, Xianyang 712100, China
2   Center for Resource Economics and Environment Management, Northwest A&F University, Xianyang 712100, China
3   College of Natural Resources and Environment, Northwest A&F University, Xianyang 712100, China
*   Correspondence: lihua7485@nwafu.edu.cn

**Abstract:** The Grassland Ecological Compensation Policy (GECP) is a key set of policy instruments designed to alleviate grassland degradation and increase herdsmen's income. However, considering the various constraints and obstacles that policies often face in actual operation, it may not be able to achieve the expected goals. In order to test the real income effect of GECP and clarify its mechanism, based on the data of 499 counties in Chinese pastoralist provinces from 2000 to 2019, this paper uses the difference-in-differences (DID) model to empirically test the impact of GECP on herdsmen's income from the dual perspective of income growth and income gap. This analysis not only evaluates the impact and mechanism of GECP on income growth in more detail, but also broadens the existing research perspective from the perspective of the income gap. The major study findings are as follows: (1) GECP significantly promotes income for herdsmen, with a marginal effect of 0.078. (2) The mechanism analysis indicates the GECP improves the income of herdsmen through the direct effect of increasing transfer income and the indirect effect of optimizing the allocation of labor, and promoting the livestock scale of barn feeding. (3) With respect to the income gap, this paper finds that areas with relatively high levels of development benefit more from GECP, which will widen the income gap between regions for herdsmen.

**Keywords:** ecological compensation; income growth; income gap; PES

## 1. Introduction

Payments for ecosystem services (PES) have received a lot of attention as an incentive-oriented environmental policy tool, especially in the intersection of poverty and ecological fragility [1,2]. PES aims at internalizing market externalities by providing compensation to the providers of ecosystem services (ES) and has gained legitimacy in developing countries due to promoting ecosystem conservation and income increase in a "win-win" manner [3,4]. In recent years, the continuous warming of the global climate has not only had a huge impact on the yield of traditional crops, but also had a huge impact on the growth of grassland, which once threatened national food security [5–7]. To protect grassland ecology and promote herdsmen's income, the Chinese government initiated a large-scale ecological compensation program, the Grassland Ecological Compensation Policy (GECP), in 2011. However, for government-financed ecological compensation programs, the buyers are not the direct users but a third party acting on behalf of the environment service users [8]. Affected by a variety of political, social, and other pressures, the tradeoff between PES's multiple goals makes people doubt PES's ability to achieve an income increase [9,10]. Therefore, it is important to correctly examine the relationship between PES and income and to clarify the internal mechanisms between them. This not only helps to consolidate and improve the grassland ecological compensation policy, but also has important practical

significance for the realization of the dual goals of grassland ecology and herdsmen's income. At the same time, it provides corresponding research methods and theoretical references for the research on the effects of other ecological compensation policies in the future.

Even though there are a lot of studies on the relationship between PES and income, the results are not all the same. Some studies argue that PES programs contribute to income improvements. For example, increased PES participation can promote income growth in China [11]. However, there is often an emphasis on the tradeoff between multiple goals in government-financed ecological compensation programs, focusing on overall regional equitability distribution [8]. As a result, fewer funds may be directed toward areas at greater risk of degradation, which are closely associated with high opportunity costs and high ecological threat activities [12]. Sims and Alis-Garcia investigated Mexico's PES scheme and found that PES has produced a significant but slight increase in poverty in areas with high participation rates [13]. In addition, Yang et al. found that the Grain-to-Bamboo Program (GTBP, which is a local PES program to grow bamboo on cropland) negatively affected income through decreased crop production [14]. In addition, due to the low quality of institutions in many developing countries, there is a high possibility of "elite capture" (elite interests becoming dominant, while the interests of disadvantaged people or those with a traditional lifestyle are ignored [15,16]) in the implementation of PES, so there is a risk of widening the income gap between different families [17]. For example, through the case of PES in China's Wolong Nature Reserve, Sheng and Wang found that PES participation in promoting income growth is more beneficial to small and medium farmers than large farmers, and this promotion effect can also exacerbate economic inequality [11].

The research specifically on the impact of GECP on income has not yet reached a consensus, and three different conclusions were formed on the significant, insignificant, and even negative effects of the policy on the herder's income increase. Hou et al. used microscopic data from 2017–2019 in five Chinese pastoralist provinces to empirically conclude that although the second round of GECP can positively promote the herder's income increase, it also exacerbates income inequality within herders [18]; Hu et al. demonstrated that the implementation of the GECP could not greatly influence the reduction in the number of cattle present on any farm size [19]; while Zhang et al. found that GECP can improve and guarantee the income level of low-income herdsmen and help narrow the gap between rich and poor through a case study approach [20]. However, the majority of the herdsmen's income level is lower than before due to insufficient relevant supporting policies. Despite the lack of consensus in the above studies, there is a growing consensus among these types of studies that GECP may not achieve the desired goals if they cannot effectively address the livelihoods of herdsmen.

Generally speaking, although the existing literature has extensively explored PES and income, there is still significant disagreement and a lack of corresponding mechanism analysis, thus providing new empirical evidence and more detailed analysis on this issue is necessary. Second, existing studies focus on the analysis of PES on income growth and are relatively inadequate on income disparity, especially in examining the differences in the average effects of PES across counties. Obviously, comprehensive consideration of income growth and the income gap can help to understand the economic effect of PES more comprehensively. To this end, in order to test the relationship between PES and income growth and income gap, and to clarify its internal mechanism, this paper uses the difference-in-differences (DID) model to empirically test the impact of GECP on herdsmen's income using county level panel data for 499 counties in six Chinese pastoralist provinces from 2000 to 2019. Then, the mediating effects model is used to empirically test the specific path of GECP on income. Finally, this paper applies the inclusive growth framework to PES programs as a means of assessing differences in how different regions benefit from GECP [21].

Compared to the research that has come before, this paper may be innovative in the following aspects: First, this study adds to the limited literature on PES and income

growth in the Chinese context [14], and examines the internal mechanism between PES and income growth through the mediating effects of transfer income, optimizing the allocation of labor, and promoting the scale of shed feeding. Secondly, this study uses the inclusive growth model to empirically test the relationship between PES and the income gap between different counties, which helps to examine the relationship between PES and income from the perspective of economic inequality [22]. Third, previous research on GECP mainly focused on Inner Mongolia, while other provinces were less involved and mainly focused on microscopic cross-sectional data [23]. However, this paper takes 499 counties in six provinces in western China as the research object, using long-term panel data from 2000 to 2019, which greatly avoids biased results caused by unobserved omitted variables.

## 2. Policy Background and Theoretical Basis

### 2.1. Study Area

The study area mainly includes six provinces: Tibet, Qinghai, Gansu, Ningxia, Inner Mongolia, and Xinjiang. It is primarily found in Northwest China. With less precipitation, the climate is temperate continental and alpine. It is an arid and semi-arid area with high terrain. The six provinces in pastoral areas have a total grassland area of 293 million hectares, respectively, accounting for 3/4 of the country's grassland area. Among them, the grassland areas of the Tibet Autonomous Region and the Inner Mongolia Autonomous Region reached 82 million hectares and 79 million hectares, accounting for 68.1% and 68.8% of the total land area of each region. The number of cattle, sheep, and large livestock in the six provinces of pastoral areas in 2020 is 38.378 million, 162.57 million, and 43.274 million heads, respectively, accounting for 40.1%, 53%, and 42.2% of the national total.

### 2.2. Policy Background

In order to prevent grassland degradation and take the livelihoods of herdsmen into account, the Chinese government enacted the "Guidance on Implementation of the Grassland Ecological Protection Subsidy and Incentive Mechanism Policy" in 2011, which marked the initial establishment of the GECP. It is also the program with the largest matching funds and the largest coverage in grassland ecological protection to date. Specifically, in the first round (2011–2015), GECP funds are mainly divided into the grassland prohibition (GP) subsidy, the grass-livestock balance (GLB) subsidy, and the production material subsidy for herding households. Among them, the GP subsidy is mainly for grasslands with poor living environments and serious grassland ecological degradation, located in big rivers, or water conservation areas, whereas the GLB subsidy is for grasslands other than the grassland prohibition area. The GLB subsidy is primarily intended to calculate the reasonable livestock carrying capacity of grasslands based on carrying capacity, and then the government will reward herdsmen for meeting requirements. According to statistics from the National Forestry and Grassland Administration, in the first round of GECP, a total of 253.4 million hectares of grassland were covered, while the subsidy funds for grassland prohibition and grass-livestock balance could reach 1.559 billion dollars per year, accounting for 81.30% of the total amount of subsidy funds.

Following the implementation of the first round of GECP, some studies reported that the low standard of compensation and laxity of regulation may be the primary reasons for ineffective programs [8]. In 2016, the Ministry of Agriculture and the Ministry of Finance of China jointly promulgated a new round of the GECP (2016–2020) for this purpose. The GP and GLB subsidies were increased from 6 to 7.5 and 1.5 to 2.5 (CNY/mu/year), respectively, in the new round of policies, while production material subsidies were eliminated. Local governments are also required to increase supervision and establish mechanisms for reward and punishment. Monitoring results show that over the 10 years following GECP implementation, total grassland vegetation cover has increased from 51% in 2011 to 56.1% in 2020, and the production of fresh grass can be as high as 1.1 billion tons. In addition, herdsmen in the policy implementation area are able to obtain 98.98 $ per capita per year, and transfer income has increased by 212 $ per household. To continue to consolidate

and enhance the achievements in grassland ecological protection, the Chinese government continues to implement the third round of GECP in 2021, increasing investment funds and further expanding the scope of implementation. The compensation area in 2021 is shown in Figure 1.

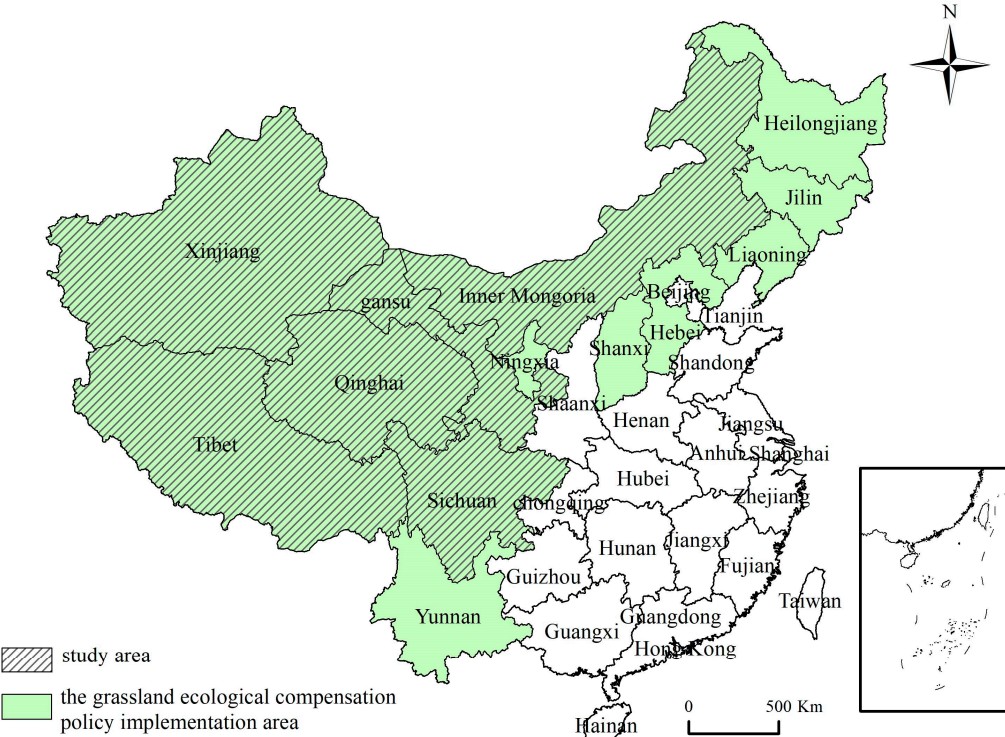

**Figure 1.** Distribution of grassland ecological compensation policy implementation provinces. The data comes from the official website of the Central People's Government of the People's Republic of China (http://www.gov.cn/zhengce/zhengceku/2022-05/13/content_5690136.htm (accessed on 15 January 2023).

It is noteworthy that in order to accelerate the dual goals of grassland ecological protection and increasing herdsmen' income, a range of supporting policies based on GECP have been issued by local governments. The program attempts to establish a modern, large-scale livestock production system in order to sustain and expand the project's implementation effect. It makes sense to speed up the transformation of herdsmen's production through subsidies to infrastructure and productive assets. They assist the herdsmen in improving the scale reward and production efficiency in order to offset losses. To this end, the government subsidized the construction of sheds and the development of artificial grass in the herding areas and adopted the "government support and self-financing of the herdsmen" method in order to encourage the herdsmen to invest in the production of herding.

### 2.3. Theoretical Basis

From a neoclassical economic perspective, the GECP will reduce herdsmen's grazing activities through both incentives and supervision, but with a substantial increase in the cost of production compared to the previous system. If herdsmen produce as an independent economic agent, they will alter the allocation of factors of production in order to maintain their returns when the quantity and price of a particular factor of production change. The fundamental question of whether GECP can achieve income growth thus lies in the trade-off between livestock producer's grazing losses and policy gains. With this in mind, this paper will conduct an in-depth analysis of the path of GECP benefits in order to explain possible changes in the herdsmen's incomes. From the above analysis, it can be seen that

the source of the herdsmen's income is divided into two levels: first, the direct effect of the GECP funds; second, the indirect effect of GECP by influencing herdsmen's production and lifestyle. Next, this paper will analyze the direct and indirect effects of GECP in detail. The framework diagram of GECP's impact on herder's income is shown in Figure 2.

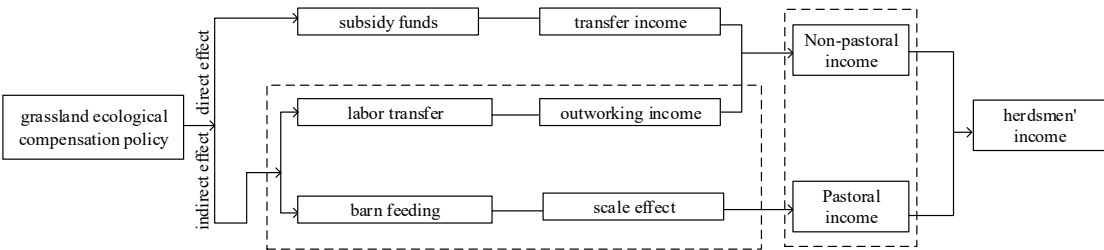

**Figure 2.** Mechanism analysis of GECP on herdsmen's income.

(1) Direct effect. The GECP will grant direct subsidy funds to herdsmen participating in the project to improve their transfer income (TI). Furthermore, subsidy funds may have heterogeneous income-increasing effects for herders with varying levels of poverty. The impact of subsidy funds on the herdsmen's income will have diminishing marginal returns as household capital increases while controlling for potential confounding factors. That means subsidy funds will have a greater marginal benefit for severely poor herdsmen than for other herdsmen in general. For example, subsidy funds account for a larger share of income for poor households and a smaller share for other households, which implies that remuneration funds have a greater marginal growth effect on poor households. At the macroeconomic level, the number of poor households is higher in economically backward areas, which results in a greater marginal benefit of grassland ecological compensation funds for backward regions than for developed ones, which in turn helps narrow the income gap between the different regions.

(2) Indirect effects. Since the implementation of the policy may alter the production and lifestyle of the herdsmen, this paper mainly analyzes the indirect effect of GECP from the two aspects of labor transfer and livestock scale of barn feeding.

For the labor transfer (LT). The GECP affects income through labor transfer in two main ways: first, the prohibition or grazing restriction reduces the labor input of grazing production and puts the liberated labor into other industries for production [24,25], increasing the time spent on non-farm or leisure [26]. Second, it enhances human capital and provides external employment opportunities [27,28]. For example, GECP can provide grassland ecological protection jobs for poor herdsmen, which in turn will have an important impact on their income [29]. In terms of impact on income gap, herdsmen in backward areas have little room for competence, which means that it is difficult for them to build labor skills that match the needs of the labor market [30]. Therefore, herdsmen in backward areas have a lower probability of obtaining either off-farm employment opportunities or higher labor returns. Consequently, there is a possibility of widening the income gap between herdsmen in different areas.

For the livestock scale of barn feeding (BF), the GECP has reduced the livestock scale of pasture feeding (PF) in the program area; herdsmen must meet the daily needs of livestock through barn feeding (BF), so the program will inevitably expand the livestock scale of BF. In terms of impact on income gap, the motivation–opportunity–ability theory posits that herdsmen are known to make feeding changes based on their knowledge and abilities. In economically developed regions, herdsmen generally have higher capacity and are more likely to switch from PF to BF with policy support, further promoting BF operations to achieve income growth. For the backward areas, the majority of herdsmen are not able to bear the huge cost of BF, so they have to reduce the number of livestock to maintain their livelihoods, which may reduce income. In turn, GECP may widen the income gap between regions by affecting livestock scale of barn feeding.

### 3. Data and Methodology

*3.1. Econometrics Model*

(1) Baseline model. The double difference in difference (DID) means that the time change of the dependent variable in the experimental group is subtracted from the time change of the dependent variable in the control group to obtain the effect of the experiment on the experimental group. "Experiment" in this paper refers to the 2011 Guidance on the Implementation of Grassland Ecological Protection Subsidy and Incentive Mechanism Policy, which the Chinese government issued and put into action. Taking 499 counties in 6 provinces in Western China as observation samples, the experimental group consisted of 332 counties that participated in GECP, and the control group consisted of 145 counties that did not participate in GECP. Therefore, in this paper, the double difference refers specifically to the changes in the per capita income of herdsmen who participated in GECP between 2000 and 2019 minus the changes in the per capita income of herdsmen in counties that did not participate in GECP to observe the impact of GECP on herdsmen's income. The time variable (time) is the dummy variable before and after the program implementation in 2011. The estimating equation to test the income growth of GECP is as follows:

$$Y_{it} = \beta_0 + \beta_1 policy * time + \theta X_{it} + a_i + \mu_t + \varepsilon_{it} \tag{1}$$

The where $i$ and $t$ represent the county and year, and $Y_{it}$ denotes the relevant indicators of income in the county $i$, including the level of rural per capita income. The coefficient $\beta_1$ of the interaction term between policy variables and time variables is the concern of this paper, which is expressed as the average treatment effect of GECP on income. $a_i$ is the individual fixed effect, $\varepsilon_{it}$ is the time fixed effect, and $\varepsilon_{it}$ is the random error term. $X_{it}$ is a set of control variables, including the degree of traditional financial development (FIN), the level of education (EDU), urbanization (URB), the level of agricultural modernization (AGR), and the degree of government participation in the economy (GOV).

(2) Dynamic effect model. This model was used to test the hypothesis of the parallel trend of DID, and to further examine the dynamic changes in the income growth of GECP. Referring to the Event Study Approach proposed by Jacobson et al., this paper constructs the following model [31]:

$$Y_{it} = \beta_0 + \sum_{t=2000}^{2019} \delta_t policy * time_{it} + \theta X_{it} + + \mu_t + \varepsilon_{it} \tag{2}$$

$\delta_t$ denotes a series of estimates from 2000–2019, and the other variables are the same as in the baseline regression.

(3) Mediating effect model. Consistent with Hayes et al. [32], the two-stage process mediation model below is proposed in order to examine whether the transfer income, labor transfer, and the livestock scale of barn feeding play an intermediary role in the process of GECP increasing the income:

$$Y_{it} = \beta_0 + \beta_1 policy * time + \theta X_{it} + a_i + \mu_t + \varepsilon_{it} \tag{3}$$

$$M_{it} = \delta_0 + \delta_1 policy * time + \theta X_{it} + a_i + \mu_t + \varepsilon_{it} \tag{4}$$

$$Y_{it} = \varphi_0 + \varphi_1 policy * time + \varphi_2 M_{it} + \theta X_{it} + a_i + \mu_t + \varepsilon_{it} \tag{5}$$

$M_{it}$ is denoted as the mediating variable, and the other variables are the same as in the baseline regression.

(4) Income gap model. To examine the impact of GECP on the income gap, this paper borrows from the inclusive growth analysis framework proposed by Zhang and Wan [21]

and incorporates the interaction term ($Y_{i,t-1} * policy$) between policy and the lagged term $Y_{i,t-1}$ in model (1), and the estimated equation is shown below:

$$Y_{it} = \beta_0 + \beta_1 policy * time + \beta_2 Y_{i,t-1} + \beta_3 Y_{i,t-1} * policy + \theta X_{it} + a_i + \mu_t + \varepsilon_{it} \quad (6)$$

When policy = 1, then:

$$E(Y_{it}|policy = 1) = \beta_0 + \beta_1 time + \beta_2 Y_{i,t-1} + \beta_3 Y_{i,t-1} + \theta X_{it} \quad (7)$$

When policy = 0, then:

$$E(Y_{it}|policy = 0) = \beta_0 + \beta_2 Y_{i,t-1} + \theta X_{it} \quad (8)$$

Thus, Equations (3) and (4) is the effect of GECP (policy) on the income ($Y_{it}$).

$$E(Y_{it}|policy = 1) - E(Y_{it}|policy = 0) = \beta_1 time + \beta_3 Y_{i,t-1} \quad (9)$$

It can be seen that the effect of GECP on income ($Y_{it}$) is divided into two parts: the first part, $\beta_1$, shows the effect of GECP on $Y_{it}$ while all other conditions do not change; the second part, $\beta_3$, is expressed as the effect of the previous period's income ($Y_{i,t-1}$) on the current period's income ($Y_{it}$) through GECP. Specifically, if $\beta_3 > 0$, counties with larger income in the previous period benefit more from GECP; on the contrary, if $\beta_3 < 0$, counties with smaller income in the previous period benefit more from GECP; furthermore, if $\beta_3 = 0$, it means that GECP does not affect the income gap. It should be noted that in estimating Equation (6), there may be endogeneity problems using the least squares method because the explanatory variables include the lagged terms of the dependent variable, so the systematic moment estimation (GMM) method of Blundell and Bond [33] is used in this paper.

### 3.2. Variable Selection

Mediating variables. In this paper, transfer income, labor transfer, and livestock scale of barn feeding were selected as mediating variables. Among them, transfer income was expressed using per capita transfer income. Labor force allocation was expressed using the annual variation of primary industry employees at the county level. The number of large-scale farms was selected as a proxy variable for barn feeding, and the greater the number of large-scale farms, the larger the livestock scale of barn feeding. In this paper, farms with an annual sheep slaughtering capacity of 100 or more heads are defined as large-scale farms according to the division of the Ministry of Agriculture and Rural Affairs of China.

Control variables. Drawing on the studies of Qi et al. [34] and Tang et al. [35], the control variables selected in this paper include (1) the degree of traditional financial development, as measured by the ratio of the loan balance of regional financial institutions to regional GDP. (2) The level of education, as measured by the proportion of illiteracy population in rural areas above 15 years old. (3) Urbanization, expressed as the proportion of total population at the end of the year relative to household population. (4) The level of agricultural modernization, expressed as the proportion of the value added of the primary sector to GDP. (5) The degree of government participation in the economy, expressed as the ratio of county-level fiscal expenditure to county-level GDP.

### 3.3. Data Description

To comprehensively examine the impact of GECP on herdsmen's income, this paper constructs 20 years panel data on county socio-economics for 499 counties in six Chinese pastoralist provinces. Specifically, firstly, considering that GECP was fully implemented in 2011, in order to obtain more diverse data to improve the accuracy and unbiasedness of the estimation results, this paper sets the sample time span as 2000–2019. On the one hand, it can satisfy the need for a parallel trend test with a double difference; on the other

hand, it can estimate the long-term dynamic effect of GECP on income. Second, due to more missing data in areas such as Qinghai Province and the Tibet Autonomous Region, the final study involved 499 counties after exclusion, including 346 in the policy group and 153 in the control group. The data are obtained from the China County (City) Social and Economic Statistical Yearbook, the China Regional Economic Statistical Yearbook, and the district and county statistical bulletins. It should be noted that all nominal variables in the data are deflated to 2000 using provincial rural CPI. Meanwhile, for some counties with missing individual indicators, this paper selects the municipal indicators where the missing counties are located for interpolation. The descriptive statistics of each variable are shown in Table 1.

**Table 1.** Descriptive statistics of the main variables.

| Variables | Observations | Mean | Min | Max | Policy Group | Control Group |
|---|---|---|---|---|---|---|
| | | | | | Mean | Mean |
| ln(income) (CNY) | 9770 | 8.426 | 6.422 | 10.392 | 8.350 | 8.597 |
| FIN (%) | 9980 | 1.856 | 0.149 | 3.312 | 1.651 | 1.454 |
| EDU (%) | 9980 | 0.167 | 0.037 | 0.491 | 0.182 | 0.155 |
| URB (%) | 7485 | 0.431 | 0.226 | 0.634 | 0.383 | 0.428 |
| AGR (%) | 9980 | 0.154 | 0.081 | 0.309 | 0.154 | 0.156 |
| GOV (%) | 9980 | 0.359 | 0.113 | 1.379 | 0.425 | 0.210 |

## 4. Empirical Results

### 4.1. Impact on Herdsmen's Income Increase: Baseline Regression Results

The average treatment effects of GECP on herdsmen's income is shown in Table 2. It can be seen that there is a positive but not significant effect of the compensation policy on herdsmen' income without controlling any variables. After adjusting for other control variables, area fixed effects, and year fixed effects in columns (2) to (4), it is discovered that GECP has a positive effect on herder's income, all of which pass the 1% significance test and the coefficients remain around 0.078. This indicates that GECP is conducive to the improvement of herdsmen's income, and this effect has certain robustness. This result is also consistent with many micro-empirical results [8,18,36]. For example, Gao et al. based on the survey data of 262 herdsmen in Inner Mongolia, found that for every 1% increase in subsidies, the income of herdsmen would increase by 0.144–0.670% [36]. At the macro level, based on the panel data of 13 provinces in China, Zhang et al. found that the implementation of GECP has reduced the income of herdsmen to a certain extent [37]. However, considering that the research focuses on the effects of policies at the national scale, and the heterogeneity among provinces is relatively strong, it may not be possible to clearly reveal the effects of policies at the county level. Meanwhile, other control variables in the model, such as the level of traditional financial development, the level of education, the urbanization rate, the level of agricultural modernization, and the degree of government participation in the economy, all have significant effects on herdsmen's income, and the explanatory power of the model even reaches 97.3%, which to a certain extent indicates the validity of the model selection.

**Table 2.** Effects of GECP on herdsmen's income: 2001–2019.

| Variables | ln(Income) (CNY) | | | |
|---|---|---|---|---|
| | **(1)** | **(2)** | **(3)** | **(4)** |
| Policy * time | 1.218 | 0.075 *** | 0.073 *** | 0.078 *** |
| | (98.758) | (9.587) | (9.341) | (13.315) |
| FIN (%) | | −0.021 *** | −0.017 *** | −0.007 *** |
| | | (−8.442) | (−7.039) | (−4.448) |
| EDU (%) | | −0.092 | −0.618 *** | 0.632 *** |
| | | (−1.090) | (−6.923) | (10.122) |
| URB (%) | | 4.870 | 4.954 | −0.089 * |
| | | (114.980) | (104.071) | (−1.747) |
| AGR (%) | | 0.035 * | 0.032 | 0.305 *** |
| | | (1.784) | (1.634) | (22.699) |
| GOV (%) | | 1.156 *** | 1.121 *** | −0.489 *** |
| | | (31.427) | (29.880) | (−16.283) |
| County fixed effects | No | No | Yes | Yes |
| Year fixed effects | No | No | No | Yes |
| Observations | 9750 | 9750 | 9750 | 9750 |
| R-squared | | | 0.927 | 0.973 |

Notes: Value out of the bracket is the parametric estimation value; value in the bracket is *t*-test value; *, *** represent significance at 10% and 1%, respectively.

*4.2. Robustness Tests*

4.2.1. Parallel Trend Test and Dynamic Effect Analysis

To test the hypothesis of the parallel trend of double difference and to further examine the dynamics of the impact of GECP on income. Figure 3 plots the estimation results of $\delta_t$ of GECP on herdsmen' income at a 95% confidence interval. Before 2011, the coefficient $\delta_t$ has been insignificant, which means that the model can well control other exogenous variables, and then satisfy the parallel trend assumption, so the double difference in difference (DID) method can be used to identify the causal effect of GECP on the income of herdsmen. Specifically, the regression coefficient between the two passed the 5% significance test after one year of the program's implementation, and the impact coefficient gradually became larger over time and then remained constant. This also indicates that the positive effect of GECP on herdsmen's income gradually becomes stronger over time and then remains stable. The explanation for this phenomenon is that, according to the program, herdsmen need to reduce the intensity of free grazing in exchange for compensation funds by changing their feeding methods or reducing the number of livestock. However, under the assumption of economic rationality, the compensation funds must be greater than the opportunity cost of herdsmen's decisions, otherwise, they will compensate for their losses by stealing grazing and night grazing. But in any case, herdsmen will always use the program to achieve an increase in their income, or at least to ensure that income does not decrease. As the program continues to advance, with the support of government measures, the cost of changing herdsmen's farming methods (pasture feeding–barn feeding) will continue to be lower, and the livestock scale of barn feeding will be expanded. The intensive and large-scale production method brought by the shed-feeding operation can greatly improve production efficiency and increase the production excess profit, thus increasing the herdsmen's income.

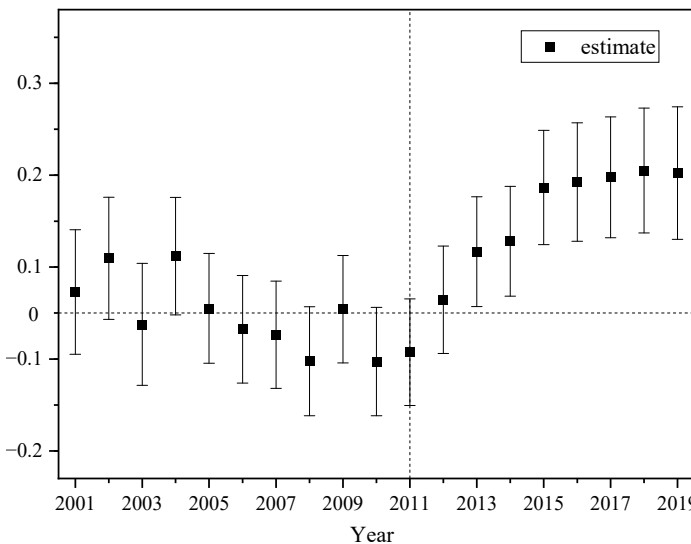

**Figure 3.** Parallel trend test.

### 4.2.2. Checking for Missing Variable Issues

In this paper, PSM-DID is used to control for individual differences between the policy and control groups. This helps to solve the possible endogeneity problem and proves the parallel trend in double difference. As shown in Table 3, after sample matching, the standardized errors of the explanatory variables are kept within 10% and the overall bias is greatly reduced. This shows that propensity score matching is a good way to deal with estimation bias caused by individual differences between groups. In the paper, the results of the matched regressions are also given. These results also show that the benchmark regressions are reliable.

**Table 3.** PSM-DID Endogeneity test.

| Variables | ln(Income)(CNY) | |
| --- | --- | --- |
| | **Coefficient Estimates** | **Standard Error** |
| Policy * time | 0.084 *** | 4.797 |
| FIN (%) | −0.010 * | −1.909 |
| EDU (%) | 1.148 *** | 0.150 |
| URB (%) | 0.003 | 0.867 |
| AGR (%) | −1.089 *** | −5.488 |
| GOV (%) | −0.529 *** | −5.189 |
| Standardization error (%) | 5.30 | |
| Observations | 8677 | |
| R-squared | 0.971 | |

Notes: Value out of the bracket is the parametric estimation value; value in the bracket is *t*-test value; *, *** represent significance at 10% and 1%, respectively.

### 4.2.3. Replacing the Explanatory Variables

In this paper, the explanatory variables were replaced to further test the validity of the benchmark regression and also to verify the effect of GECP on herdsmen's income. Specifically, county GDP per capita was used instead of herdsmen per capita income, because although GDP per capita represents local economic development, it can also reflect local income level. Based on this, a benchmark regression model was used to conduct a linear regression of the impact of GECP on per capita GDP, and from the empirical results in column (1) of Table 4, the coefficient of GECP was significantly positive after replacing the explanatory variables, indicating that the main findings of this paper remain robust.

**Table 4.** Robustness test.

| Variables | (1) | (2) | (3) |
|---|---|---|---|
| | ln(GDP) (CNY) | ln(Income) (CNY) | ln(Income) (CNY) |
| Policy * time | 0.061 ** | 0.079 *** | 0.078 *** |
| | (2.324) | (10.150) | (4.627) |
| Control variables | Yes | Yes | Yes |
| County fixed effects | Yes | Yes | Yes |
| Year fixed effects | Yes | Yes | Yes |
| Observations | 7704 | 9261 | 9750 |
| R-squared | 0.950 | | 0.975 |

Notes: Value out of the bracket is the parametric estimation value; value in the bracket is *t*-test value; **, *** represent significance at 5% and 1%, respectively.

### 4.2.4. Dealing with Reverse Cause and Effect

Considering that the explanatory variables and control variables may be in the presence of reverse causality, this paper lags all control variables by one period before regression estimation, in order to solve the possible endogeneity problem. The estimated results are shown in column (2) of Table 4, and the herdsmen's income is basically consistent with the results of the benchmark regression, which all show a significant positive correlation, thus verifying the robustness of the benchmark regression model.

### 4.2.5. Adding City and Year Fixed Effects

Although this paper controls for county and year-fixed effects in the benchmark regression model, herdsmen's income may also be affected by the characteristic factors of the city where they are located as well as event shocks, so this paper further incorporates city and year interaction fixed effects into the benchmark regression model to control for the effects of time-varying characteristics or shocks at the city level. Column (3) in Table 4 reports the regression results, and it can be found that the effect of GECP on herdsmen's income remains largely consistent with the benchmark regression, which again verifies the robustness of the benchmark regression.

### 4.2.6. Placebo Test

Given the possibility of unobservable factors other than GECP influencing income, which could lead to biased estimation results. In this paper, a placebo test and a double difference to construct a virtual environment with a quasi-natural experiment are used to test for possible risks. Specifically, a random sampling method was used to generate a random experimental group for the annual GECP, which in turn generated estimates of the coefficients of the wrong multiplicative difference term. A total of 1000 random samples were conducted in this paper, and the samples from each sample were included in a double-difference benchmark regression model for estimation. If the constructed independent variables do not have a significant effect on herdsmen's income, then the baseline regression results can be said to be more robust. Figure 4 shows the distribution of the *t*-values of the regression coefficients of pastoralist per capita income after 1000 samples. It can be seen that the *t*-values basically accept the original hypothesis, which indicates that the baseline regression results of this paper remain robust after excluding other unobservable factors.

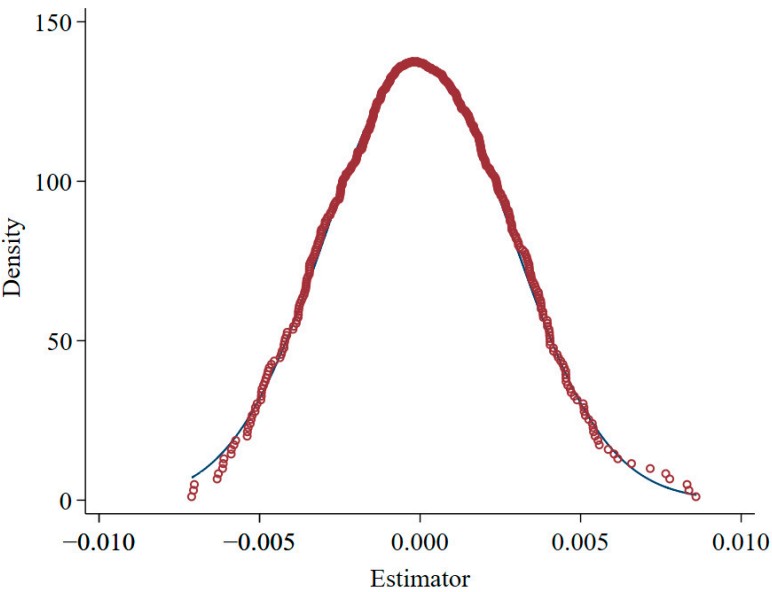

**Figure 4.** Placebo test.

*4.3. Mechanisms Analysis*

The previous analysis shows that GECP has a statistically significant contribution to herdsmen's income at the statistical level. However, a more intuitive economic explanation of the mechanism of this impact path is needed. To this end, based on the above theoretical analysis, this paper constructs a mediating effect model to verify the mechanism of the effect of GECP on herdsmen's income one by one.

A step-by-step test is used to examine the intermediary effects of transfer income, labor transfer, and livestock scale of barn feeding. In addition to this, Sobel and Bootstrap methods are applied to again test the mediation effect in order to enhance the robustness of the results. The results of the intermediary effect are populated in Table 5.

**Table 5.** Test of mediating effect of GECP on income.

| Variables | Transfer Income | | Labor Transfer | | Livestock Scale of Barn Feeding | |
|---|---|---|---|---|---|---|
| | (1) | (2) | (3) | (4) | (5) | (6) |
| | ln(TI) | ln(income) | ln(LT) | ln(income) | ln(BF) | ln(income) |
| Policy * time | 0.030 ** | 0.073 *** | −11.107 *** | 0.076 *** | 0.344 *** | 0.056 *** |
| | (2.383) | (4.409) | (−11.033) | (4.659) | (7.962) | (3.012) |
| ln(TI) (CNY) | | 0.107 *** | | | | |
| | | (16.589) | | | | |
| ln(LT) (people) | | | | 0.001 *** | | |
| | | | | (8.215) | | |
| ln(BF) (household) | | | | | | 0.255 *** |
| | | | | | | (20.402) |
| Control variables | Yes | Yes | Yes | Yes | Yes | Yes |
| Fixed effects | Yes | Yes | Yes | Yes | Yes | Yes |
| R-squared | 0.981 | 0.974 | 0.251 | 0.973 | 0.513 | 0.827 |
| Sobel test (Z) | 22.71 *** | | −8.065 *** | | 17.51 *** | |
| Bootstrap test | [0.005, 0.043] | | [−0.026, −0.016] | | [0.574, 0.638] | |

Notes: Value out of the bracket is the parametric estimation value; value in the bracket is *t*-test value; **, *** represent significance at 5% and 1%, respectively.

There are three different situations based on the mediation effect model. The details for each situation are outlined below: First, the mediating effect of the transfer income. The regression coefficient of column (1) in Table 5 is positive (0.030) at the 1% significance level. The means by which GECP causes the transfer income to increase. Furthermore,

column (2) of Table 5 shows that the coefficients of the GECP (0.073) and transfer income (0.107) are also significant at 1%; thereby, confirming a mediation effect. Secondly, the mediating effect of the labor transfer. The regression coefficient of column (3) in Table 5 is negative (−11.107) at the 1% significance level. This implies that GECP can promote labor migration. The coefficients of these two variables in column (4) are significant and positive at the 1% significance level. This shows that GECP can accelerate labor transfer, thereby increasing herdsmen's income. However, this result is controversial in existing studies. Some studies believe that participating in ecological compensation plans will reduce dependence on natural resources, and then encourage labor to migrate to increase non-agricultural income [38,39]. However, this process may be affected by personal factors, such as age and education level, which makes the effect of policy on non-agricultural activities less significant [40,41]. However, this paper argues that with the continuous increase in the implementation of PES, in order to ensure that the livelihood level does not decline, herdsmen will break through the relevant constraints and actively go out to work to increase their income. Third, the mediating effect of the livestock scale of barn feeding. The coefficients of these two variables in both columns (5) and (6) are significant and positive at the 1% significance level. This shows that GECP promotes income increase through the livestock scale of barn feeding.

Lastly, the 95% confidence interval of the Boostrap test does not include 0, and the value of the Sobel test is also significant. This infers that the results of the mediation effect are reliable.

*4.4. Impact on Herdsmen's Income Gap*

The above empirical evidence has shown the impact of GECP on herdsmen's income growth and its mechanism of action, but does this income growth effect have a heterogeneous impact on different regions? In other words, how is the program's growth effect distributed across regions, and whether regions lagging behind in economic development benefit more from it, or whether it is more conducive to improving the economic level in developed regions, thus widening the regional development gap? The answer to this question can help grasp the effect of GECP at a global level, but there is little literature to explore this issue.

To this end, this paper draws on the empirical model of inclusive growth proposed by Zhang et al. [21], Equation (6), to test the impact of GECP on the income gap. In order to avoid bias in the estimation results, this paper uses the systematic GMM method. Among them, the GMM-type variable is the lagged term of herdsmen's income and the interaction term with policy variables. Meanwhile, the lag order is determined according to whether the model satisfies the assumptions of autocorrelation and over-identification. It should be noted that $\beta_3$ is the coefficient we are most interested in. If $\beta_3$ is significantly negative, it indicates that the relatively backward counties can benefit more from the policy, and conversely, the economically developed counties are more likely to profit from the policy, which means GECP widens the income gap between different regions.

Table 6 reports the regression results on Equation (6), and it can be seen that the systematic GMM estimation results of GECP on herdsmen's income pass the corresponding autocorrelation test and over-identification test, indicating that the model setting is not significantly biased. From the regression coefficients, it is found that the interaction term between GECP and farmers' income is significantly positive, $\beta_3 > 0$. This indicates that counties with higher herdsmen's income benefit more from GECP, and therefore, the policy widens the income gap between herdsmen in different regions. Compared with previous research results, based on survey data of 203 herdsmen households in Xin Barag Left Banner, Inner Mongolia Autonomous Region, Li et al. showed that GECP also widened the absolute income gap among herdsmen households [42]. The above shows that after the implementation of the GECP, due to the significant differences in the difficulty of changing the breeding mode and the choice of non-agricultural employment opportunities

for herdsmen with different incomes, there are also significant differences in the response to the policy, which will also increase income gap among different herdsmen.

**Table 6.** Regression results of GECP on income gap.

| Variables | ln(Income) (CNY) | |
| --- | --- | --- |
| | Coefficient Estimates | Standard Error |
| Policy | 0.346 *** | 8.749 |
| L. ln(income) * Policy | 0.033 * | 1.844 |
| L. ln(income) (CNY) | 0.245 *** | 4.655 |
| FIN (%) | −0.089 *** | −7.488 |
| EDU (%) | 1.012 *** | 0.212 |
| URB (%) | 3.987 *** | 13.328 |
| AGR (%) | −0.341 *** | −5.537 |
| GOV (%) | 1.338 *** | 12.975 |
| AR(2)-P | 0.173 | |
| Hansen-p | 0.192 | |
| Observations | 9246 | |

Notes: Value out of the bracket is the parametric estimation value; value in the bracket is *t*-test value; *, *** represent significance at 10% and 1%, respectively.

## 5. Conclusions

Considering the profound impact of continuous climate change on the grassland ecosystem [5], we believe that it is necessary to test the implementation effect of GECP from a larger time and space dimension, so as to provide suggestions for improving the policy. The grassland ecological compensation policy is one of the most important ways that China protects the environment. It aims to drive the transformation and upgrading of grassland livestock production and operations, improve herdsmen's income, and promote the protection and improvement of grassland ecology with the help of performance assessment incentive funds and related supporting policies. Therefore, a more comprehensive assessment of GECP is important not only for understanding the performance of grassland ecological compensation in China, but also for enriching the general understanding of the effects of ecological compensation policies.

The GECP covers the major pastoral areas of China, and the large scope of policy implementation provides a valuable opportunity to identify the effects of large-scale ecological compensation policies. Based on county-level panel data from 2000 to 2019 for 499 counties in six provinces in China's pastoral areas, this paper assesses the impact of GECP on income growth and the inter-regional income gap, the latter providing a new perspective for ecological compensation policy evaluation. The study found that in terms of income growth, GECP can significantly increase herdsmen's income, and the intensity of the effect on income growth gradually becomes stronger in the first 5 years of the policy and then flattens out; the mechanism analysis shows that the policy mainly achieves herdsmen's income growth through the direct effect of compensation funds and the indirect effect of transferring labor and improving the livestock scale of barn feeding. In terms of the income gap, counties with relatively higher per capita income levels of herdsmen benefited more from the policy, indicating that the policy widened the development gap within regions.

The study in this paper responds to the research that has already been done on the different effects of ecological compensation policies on income. It also gives strong evidence that ecological compensation policies have a positive effect on income. On the one hand, this paper argues that GECP significantly increases the average income of herdsmen in the policy implementation area. The mechanism analysis shows that the policy can promote the transformation and upgrading of herdsmen's production and lives through the paths of influencing compensation funds, labor distribution, and the scale of bred feeding, and thus increase income. On the other hand, this paper finds that the effectiveness of these paths is heterogeneous across different economic development regions, which may

widen the income gap between regions. Herdsmen in lagging regions may not be able to afford the high cost of production transformation, and the policy implementation may be further exploited by the non-compliant use of funds and "elite capture". As a result, the policy should pay attention to and increase support for herder production transformation in backward areas, effectively reduce the costs and risks of herder transformation, and ultimately achieve inclusive income growth for herdsmen. Of course, there are some shortcomings in this study. For example, although this study has grasped the overall effect of GECP on herdsmen's income from the county level, the responses to the policy may vary in different regions. The heterogeneity of income has not been analyzed empirically, and the formation mechanism of heterogeneity has not been discussed in depth. This will also be one of the directions that follow-up research needs to focus on.

**Author Contributions:** Conceptualization, M.L. and L.B.; methodology, M.L. and L.B.; software, M.L. and L.B.; validation, M.L. and H.L.; formal analysis, M.L.; investigation, M.L.; resources, M.L.; data curation, M.L.; writing—original draft preparation, M.L. and H.S.K.; writing—review and editing, M.L. and H.S.K.; visualization, M.L.; supervision, H.L.; project administration, H.L.; funding acquisition, H.L. All authors have read and agreed to the published version of the manuscript.

**Funding:** This research was funded by the Key think tank research projects on major theoretical and practical issues in philosophy and social sciences in Shaanxi Province: value realization mechanism of ecological products and value realization path design of typical ecological products in Shaanxi (2021ZD1041).

**Institutional Review Board Statement:** Not applicable.

**Informed Consent Statement:** Not applicable.

**Data Availability Statement:** The data presented in this study are available in this article.

**Conflicts of Interest:** The authors declare no conflict of interest.

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
