# Peer review of "The Influence of the Grassland Ecological Compensation Policy on Regional Herdsmen’s Income and Its Gap: Evidence from Six Pastoralist Provinces in China"

_agriculture, doi:10.3390/agriculture13040775_

Round 1

Reviewer 1 Report

1. The abstract lacks the purpose, relevant information about the sources of data and the timeframe of the research.

2. Although the justification for undertaking the research is relatively detailed, at the end of the introduction it is indicated what the paper contains and what it draws attention to, there is no clear indication of the main purpose of the research, which makes it difficult to evaluate it, especially in the layer of formulated conclusions.

3. There is no reference in the text to graphic elements, for what year are the data presented in Figure 1?; the title of the Figure 2 to be corrected.

4. The layout of Chapter 2.2 is unclear and requires reorganization. There are unclear / illegible phrases in the following paragraphs, e.g. Line 171 For the labor allocation (LA); line 192 For the barn feeding (BF) scale. The organization of this theoretical part needs to be sorted out.

5. Definitely, the weakness of the manuscript is the lack of discussion of research results with the work of other authors. This is also reflected in the relatively short list of research papers used.

6. The limitations of the conducted research should be emphasized.

Author Response

Point 1: The abstract lacks the purpose, relevant information about the sources of data and the timeframe of the research.

Response 1: Thanks for the reviewer’s kind suggestion. We have made correction according to the reviewer’s Points. The revised details can be found in the red font section on lines 14-22.

Point 2: Although the justification for undertaking the research is relatively detailed, at the end of the introduction it is indicated what the paper contains and what it draws attention to, there is no clear indication of the main purpose of the research, which makes it difficult to evaluate it, especially in the layer of formulated conclusions.

Response 2: Thanks for the reviewer’s kind suggestion. We have made correction according to the reviewer’s Points. The revised details can be found in the red font section on lines 45-50, lines 91-92.

Point 3: There is no reference in the text to graphic elements, for what year are the data presented in Figure 1?; the title of the Figure 2 to be corrected.

Response 3: Thanks for the reviewer’s kind suggestion. We have made correction according to the reviewer’s Points. The revised details can be found in the red font section on page 4 and page 5.

Point 4: The layout of Chapter 2.2 is unclear and requires reorganization. There are unclear / illegible phrases in the following paragraphs, e.g. Line 171 For the labor allocation (LA); line 192 For the barn feeding (BF) scale. The organization of this theoretical part needs to be sorted out.

Response 4: Thanks for the reviewer’s kind suggestion. We have made correction according to the reviewer’s Points. The revised details can be found in the red font section on pages 5-6.

Point 5: Definitely, the weakness of the manuscript is the lack of discussion of research results with the work of other authors. This is also reflected in the relatively short list of research papers used.

Response 5: Thanks for the reviewer’s kind suggestion. We have made correction according to the reviewer’s Points. The revised details can be found in the red font section on lines 391-399, lines 520-528, lines 563-570.

Point 6: The limitations of the conducted research should be emphasized.

Response 6: Thanks for the reviewer’s kind suggestion. We have made correction according to the reviewer’s Points. The revised details can be found in the red font section on lines 608-613.

Reviewer 2 Report

Dear authors, I congratulate you on your choice of research topic. It is important both for the dynamics of scientific research and for practical use. However, I think some small additions are necessary:

The abstract could be more detailed to increase the visibility of the paper. Also, more keywords with the same purpose can be indicated.

The abstract should also include the new contribution of the research.

Detail the state of research in the field investigated by the article. Add other references.

The introduction could be completed by checking the following components:

1.            Social and economic significance and justification of research results,

2.            Importance and necessity of the intention to undertake in this field,

3.            Statement of the objective at the end of the introduction, as well as the method used and the new contributions of the study. The more obvious novelty of the research presented in lines 73-84: "Compared to the research that has come before, this paper may only add a little bit 73 in three ways:"

Also, in the introduction, presenting the characteristics of the area and population researched.

In section "2.1. Policy background" I am not sure if the source for Figure 1. Distribution of grassland ecological compensation policy implementation provinces was specified.

You can specify the source of the information "Statistics show that in the first round of GECP, a total of 253.4 million hectares of grassland were covered, while the subsidy funds for grassland prohibition and grass-livestock balance could reach 1.559 billion dollars per year, accounting for 81.30% of the total amount of subsidy funds.".

In the research methodology, the difference-in-differences (DID) model hypothesis should be highlighted. In addition, it would be useful to detail the relationship between the comparison groups.

Place more emphasis on the conformity or differences of your own results with those of other research with the same objectives.

I did not identify in the article the limitations of the research and recommendations for future research. If there are none, it would be good to make these points.

Finally, I believe that the bibliography can be supplemented with articles relevant to this research.

I congratulate you on your valuable research and wish you success!

Author Response

Point 1: The abstract could be more detailed to increase the visibility of the paper. Also, more keywords with the same purpose can be indicated.

Response 1: Thanks for the reviewer’s kind suggestion. We have made correction according to the reviewer’s Points. The revised details can be found in the red font section on page 1.

Point 2: The abstract should also include the new contribution of the research.

Response 2: Thanks for the reviewer’s kind suggestion. We have made correction according to the reviewer’s Points. The revised details can be found in the red font section on lines 20-22.

Point 3: Detail the state of research in the field investigated by the article. Add other references.

Response 3: Thanks for the reviewer’s kind suggestion. We have made correction according to the reviewer’s Points. The revised details can be found in the red font section on lines 70-82.

Point 4: The introduction could be completed by checking the following components:

  1. Social and economic significance and justification of research results,

  1. Importance and necessity of the intention to undertake in this field,

  1. Statement of the objective at the end of the introduction, as well as the method used and the new contributions of the study. The more obvious novelty of the research presented in lines 73-84: "Compared to the research that has come before, this paper may only add a little bit 73 in three ways:"

Also, in the introduction, presenting the characteristics of the area and population researched.

Response 4: Thanks for the reviewer’s kind suggestion. We have made correction according to the reviewer’s Points. The revised details can be found in the red font section on lines 45-49, lines 91-92, lines 99-100. At the same time, The regional and demographic characteristics of the study are briefly mentioned in the abstract and detailed in 2.1.

Point 5: In section "2.1. Policy background" I am not sure if the source for Figure 1. Distribution of grassland ecological compensation policy implementation provinces was specified.

Response 5: Thanks for the reviewer’s kind suggestion. We have made correction according to the reviewer’s Points. The revised details can be found in the red font section on page 4, and details of grassland ecological compensation policy implementation provinces can be found on lines 113-123.

Point 6: You can specify the source of the information "Statistics show that in the first round of GECP, a total of 253.4 million hectares of grassland were covered, while the subsidy funds for grassland prohibition and grass-livestock balance could reach 1.559 billion dollars per year, accounting for 81.30% of the total amount of subsidy funds.".

Response 6: Thanks for the reviewer’s kind suggestion. We have made correction according to the reviewer’s Points. The revised details can be found in the red font section on lines 137-138.

Point 7: In the research methodology, the difference-in-differences (DID) model hypothesis should be highlighted. In addition, it would be useful to detail the relationship between the comparison groups.

Response 7: Thanks for the reviewer’s kind suggestion. We have made correction according to the reviewer’s Points. The revised details can be found in the red font section on lines 275-287.

Point 8: Place more emphasis on the conformity or differences of your own results with those of other research with the same objectives.

Response 8: Thanks for the reviewer’s kind suggestion. We have made correction according to the reviewer’s Points. The revised details can be found in the red font section on lines 391-399, lines 520-528, lines 563-570.

Point 9: I did not identify in the article the limitations of the research and recommendations for future research. If there are none, it would be good to make these points.

Response 9: Thanks for the reviewer’s kind suggestion. We have made correction according to the reviewer’s Points. The revised details can be found in the red font section on lines 608-613.

Reviewer 3 Report

In this paper, the authors have investigated the influence of grassland ecological compensation policy on regional herdsmen' income and its gap using DID model for the county-level panel data. The following are the comments that need the kind attention of the authors for improvement.

·       The title and abstract are good and shall be retained.  

·       L67-72: This has to be given in the data and methodology section.

·       L231-232, 323-324.: The sentence is not continuous

·       Why have the authors applied both fixed effects and random effects models?

·       Table 1: Explanation for variables (first column) along with the units have to be presented. Why do the variables’ frequency differ across regions?

·      Any sensitivity analysis done to estimate the reliability of the given models?

·       In-text citations are missing in the R&D section.

·       There are some grammatical and punctuation errors in the manuscript. The authors are suggested to fix such errors or avail of the services of proof editing.

·       The conclusions and recommendations should emerge only from the study. Also, reference books are given on the agriculture web. This will help you to take a call on the paper.

Author Response

Point 1: L67-72: This has to be given in the data and methodology section.

Response 1: Thanks for the reviewer’s kind suggestion. We have made correction according to the reviewer’s Points. The revised details can be found in the red font section on lines 275-287.

Point 2: L231-232, 323-324.: The sentence is not continuous.

Response 2: Thanks for the reviewer’s kind suggestion. We have made correction according to the reviewer’s Points. The revised details can be found in the red font section on line 309-311, line 412-416.

Point 3: Why have the authors applied both fixed effects and random effects models?

Response 3: Thanks for the reviewer’s kind suggestion. This article does not use a random effect model, but chooses to use a time and individual double fixed effect model. If a random effect model needs to be added as a robustness test, this article will add corrections later.

Point 4: Table 1: Explanation for variables (first column) along with the units have to be presented. Why do the variables’ frequency differ across regions?

Response 4: Thanks for the reviewer’s kind suggestion. We have made correction according to the reviewer’s Points. The revised details can be found in the red font section on lines 382-383. It should be noted that the data in this paper mainly come from national and local statistical yearbooks. Due to the lack of some statistical yearbooks in some backward areas, some indicators in this area are missing, so that the variables’frequency differ across regions.

Point 5: Any sensitivity analysis done to estimate the reliability of the given models?

Response 5: Thanks for the reviewer’s kind suggestion. Section 4.2 is the sensitivity test for the baseline regression model in this paper. If additional additions are required, this article will be amended later.

Point 6: In-text citations are missing in the R&D section.

Response 6: Thanks for the reviewer’s kind suggestion. We have made correction according to the reviewer’s Points and missing literature has been added. The revised details can be found in the red font section on lines 689-702.

Point 7: There are some grammatical and punctuation errors in the manuscript. The authors are suggested to fix such errors or avail of the services of proof editing.

Response 7: Thanks for the reviewer’s kind suggestion. We has corrected some grammatical and punctuation errors, and at the same time tried to communicate with the editor to further complete the proofreading.

Point 8: The conclusions and recommendations should emerge only from the study. Also, reference books are given on the agriculture web. This will help you to take a call on the paper.

Response 8: Thanks for the reviewer’s kind suggestion. We have made correction according to the reviewer’s Points, pointing out the deficiencies of the article. The revised details can be found in the red font section on lines 608-613.

Round 2

Reviewer 3 Report

Dear Authors,

Thank you for making a good attempt at addressing the comments. I am satisfied with a majority of the responses and commend the authors on their effort. Some minor errors need to be fixed before making the final decision by the Editor on the manuscript. I hope the outcome of this review process brings you some joy. Best wishes!

·        Line 105: ‘marginal’ word has to be deleted

·        Units have to be given for the variable names (multiple tables)

Author Response

Point 1: Line 105: ‘marginal’ word has to be deleted.

Response 1: Thanks for the reviewer’s kind suggestion. We have made correction according to the reviewer’s Points. The revised details can be found in the red font section on lines 99-100.

Point 2: Units have to be given for the variable names (multiple tables)

Response 2: Thanks for the reviewer’s kind suggestion. We have made correction according to the reviewer’s Points. We have added units to all variables in the table.
